# Vascular Distribution and Expression Patterns of Angiogenic Factors in Caruncle during the Early Stage of Pregnancy in Goats (*Capra hircus*)

**DOI:** 10.3390/ani13010099

**Published:** 2022-12-27

**Authors:** Pan Wang, Nanjian Luo, Le Zhao, Yongju Zhao

**Affiliations:** Chongqing Key Laboratory of Herbivore Science, Chongqing Key Laboratory of Forage & Herbivore, Chongqing Engineering Research Center for Herbivores Resource Protection and Utilization, College of Animal Science and Technology, Southwest University, Chongqing 400715, China

**Keywords:** goat, caruncle, angiogenesis, early pregnancy, DNA methylation

## Abstract

**Simple Summary:**

Placental blood circulation is the carrier of material exchange, which affects successful pregnancy and fetal development. Placental angiogenesis is very important for the embryo implantation, normal growth and development of the fetus, and animal reproductive performance, but its regulatory mechanism is very complex. In this study, we investigated the relationship between vascular distribution and the expression and methylation patterns of major angiogenic factors in the goat caruncle during the early stage of pregnancy. This research aimed to better understand the genetic mechanism of animal reproductive performance differences and the molecular mechanism of caruncle angiogenesis.

**Abstract:**

The placenta is a temporary maternal–fetal organ, and its maternal placenta (caruncle) is essential for fetal growth and development. The exchange function of the placenta requires vascular development (angiogenesis). However, the angiogenesis of the caruncle is poorly understood in goats during the early stage of pregnancy. Here, we investigated the vascular distribution, mRNA expression of major angiogenic factors, and the methylation levels of *ANGPT2* in the goat caruncle. It showed that CAD (capillary area density), CSD (capillary surface density), and APC (area per capillary) increased gradually, while CND (capillary number density) showed an insignificant change, probably due to the variability between animals. The proportion of proliferating cells was observed to be very high (>26%) and increased (*p* < 0.002) approximately 2-fold from day 20 to 60 of pregnancy. Furthermore, the expression patterns of major angiogenic factors changed during the early stage of pregnancy. Interestingly, we discovered an absolute correlation between the mRNA for *ANGPT2*, *TEK*, *FGF2,* and vascular distribution. Subsequently, we evaluated the DNA methylation of *ANGPT2*, where we found that mean methylation was negatively correlated with CAD. The methylation at the CpG sites, such as CpG 4/18, CpG 9.10.11, and CpG 15, showed significant changes during the early stage of pregnancy. Thus, our findings suggest that the methylation of *ANGPT2* may be involved in the regulation of caruncle angiogenesis during the early stage of pregnancy.

## 1. Introduction

Placenta, containing the maternal component (caruncle) and the fetal component (cotyledon), has a vital role in fetal development during pregnancy in mammals. Interestingly, maternal caruncles exist on the endometrial surface throughout the life of the female and serve as a connecting medium between the endometrium and fetal membranes [1]. As the pregnancy progresses, the size, total volume, and the surface area of the placentome (containing both the maternal caruncular and fetal cotyledonary components) increase gradually, thus, enhancing the maternal–fetal contact area [1,2]. However, the vascular network between the maternal and the fetal system is essential for maintaining placental function and growth, and meeting the needs of fetal development.

Placental angiogenesis begins in the early stage of pregnancy, and the increased vascularization occurs through the coordinated growth of the vascular network in the uterus [3]. In humans, the initiation of the extensive angiogenic remodeling of the endometrium is detected on gestation day 21 [4]; in sheep, microvilli vessels are observed on gestation day 18 [5]; and in rats, the proliferation of endothelial and perivascular cells begins to increase on gestation day 3 [6]. Normal angiogenesis is essential for fetal growth and development. Several aspects of placental function are involved in the regulation of angiogenesis, such as the expression of various factors, methylation patterns, and other processes [7]. For example, the vascular endothelial growth factor (VEGF) and angiopoietins (ANGPT) family are two specific signaling pathways, which regulate vascular endothelial cells and play a key role in angiogenesis [8,9,10,11]. VEGFA, one of the main angiogenic factors, promotes angiogenesis by binding to its receptor KDR, while another receptor FLT1 has an inhibitory effect via suppressing the KDR signal [12]. The ANGPT1/2-TEK is another pathway regulating the integrity of the blood vessel wall [13]. In humans, ANGPT1 is a vascular stabilizer acting on the TEK receptor, whereas ANGPT2 is a negative regulator that disrupts the blood vessel homeostasis [14,15]. Fibroblast growth factor 2 (FGF2), a member of the fibroblast family, provides the primitive stimulation signals for initiating angiogenesis, and thus contributes to the vascular endothelial cell proliferation and angiogenesis [16,17,18]. During the early stage of pregnancy in rats, sheep, and rhesus monkeys, *VEGFA*, *ANGPT1/2*, *FGF2* and their receptors are detected, and these participate in the regulation of angiogenesis in placenta [6,19,20]. 

Furthermore, DNA methylation is a well-known mechanism that is involved in embryonic development and fetal growth [21]. Recent studies have suggested that the abnormal methylation of angiogenic factors might cause inadequate placental angiogenesis, leading to a gradual decline in placental function and limited fetal intrauterine growth; this results in abortion, premature birth, or both maternal and fetal death [7,21,22,23]. These observations suggest that the adequate development of placental vasculature is extremely conducive to intrauterine fetal development. However, the expression pattern of angiogenic factors and the placental vascular distribution in goats remain unclear. Thus, placental angiogenesis in goats needs to be investigated to understand the mechanisms involved in embryonic growth, thus improving their reproductive performance.

In this study, we observed a gradual increase in the vascular distribution and the proliferation of endothelial and perivascular cells in the caruncle during the early stage of pregnancy. We discovered that the mRNA expression of *ANGPT2*, *TEK*, and *FGF2* was correlated with the vascular distribution. Furthermore, we explored the DNA methylation of *ANGPT2*, which was also correlated with the vascular distribution. There were significant changes in the mean methylation of the *ANGPT2* and several CpG sites during the early stage of pregnancy. Our findings indicate that the methylation of *ANGPT2* was involved in the regulation of caruncle angiogenesis during the early stage of pregnancy.

## 2. Materials and Methods

### 2.1. Animals and Caruncle Collection

All animal procedures were approved by the Southwest University Institutional Animal Care and Use Committee (2019, No. GB14925-2010). Fifteen 8-month-old Dazu black female goats were supplied by Dazu Black Goat Breeding Farm and raised at Southwest University. These goats were virgin and weighed about 25–30 kg; they had an unrestricted access to straw, water, and mineral salt lick. They were randomly divided into five groups (*n* = 3 for each group) and naturally mated with a 2-year-old Dazu Black buck. The caruncle tissues were excised from the gravid uterine horn closest to the embryo on gestation day 20, 25, 30, 45, and 60 after slaughter (day of mating = day 0), and stored at -80 °C until used for RNA/DNA extraction. For immunohistochemical staining, the whole caruncle was fixed in 4% paraformaldehyde (PFA) and stored at 4 °C.

### 2.2. Immunohistochemistry and Image Analysis

The whole caruncle samples were fixed in 4% PFA overnight and dehydrated in 70–100% ethanol, embedded in paraffin, blocked, sectioned (3 μm), and then incubated with anti-CD31 (blood vessel marker; 1:300; rabbit polyclonal antibody; Beijing Biosynthesis Biotechnology Co., Ltd., Beijing, China) and anti-Ki67 (cell proliferation marker; 1:400; rabbit polyclonal antibody; Beijing Biosynthesis Biotechnology Co., Ltd., Beijing, China) for vascular examinations. Briefly, deparaffinized tissue sections were treated with heat-induced antigen retrieval in 0.01 M citrate buffer, followed by blocking with 3% BSA for 30 min at room temperature. Blocked sections were incubated with an anti-CD31 antibody or anti-Ki67 antibody overnight at 4 °C. After incubation with a secondary antibody, sections were developed using 3,3-diaminobenzidine (DAB; Servicebio, Wuhan, China). All specimens were counterstained with hematoxylin.

For each tissue section, images of 2–3 randomly chosen fields from caruncle per animal (total areas ≥ 1 × 10^7^ μm^2^) [1] were taken at 100× magnification (CD31 staining), using Sunny Digital microscope lab system computer software (Ningbo Sunny Instruments Co., Ltd., Ningbo, China) to determine vascularity, including CAD (total capillary area/tissue area), CSD (total capillary circumference/tissue area), CND (total number of capillaries/tissue area), and APC (area per capillary). Ki67 was evaluated as percentages of labeled endothelial and perivascular cells. For calculating the % of Ki67, 15–20 randomly chosen fields were recorded at 400× magnification per animal [5] and expressed as a percentage of positive cells.

### 2.3. RNA Isolation and Quantitative Real-Time RT-PCR for mRNA Expression

To detect the mRNA expression of major angiogenic factors, approximately 50 mg of tissue from the center of a complete caruncle were taken and homogenized with ice-cold TRIzol (Ambion, Foster City, CA, USA). Total RNA extraction was performed according to the manufacturer’s instructions. The first strand cDNA was synthesized from 1 µg total RNA using Fast Quant RT Kit (with gDNA) kit (TIANGEN Biotech, Beijing, China). The gene-specific primer sequences for amplification of the angiogenic factors (*VEGFA*, *FLT1*, *KDR*, *ANGPT1*, *ANGPT2*, *FGF2* and *FGFR2*) and internal control (β-actin), are listed in Table 1. The mRNA expression levels were measured by quantitative real-time PCR, and the reactions were performed in a C1000-CFX96 Real-Time PCR System (Bio-Rad, San Francisco, CA, USA) using SYBR Green Master Mix (Applied Biosystems, Foster City, CA, USA) according to the manufacturer’s recommendations. For each sample, three replicates were performed under the following program: 95 °C for 10 min; 39 cycles for 15 s at 95 °C; and 59 °C for 1 min. The relative transcript abundance of the angiogenic factors was normalized to the β-actin and calculated using the 2^−ΔΔ*C*T^ method.

### 2.4. ANGPT2 Methylation Assay

For *ANGPT2* methylation analysis, approximately 50 mg of caruncle sample was taken and isolated for genomic DNA using the TIANamp Genomic DNA Kit (TIANGEN Biotech, Beijing, China). We used the Sequenom MassARRAY EpiTYPER approach (Sequenom, San Diego, CA, USA) to measure the DNA methylation of CpG sites in the 5′ UTR region of the *ANGPT2* gene. The primers (forward: aggaagagagGGAGTAAGTATTTTAGGTTTTTGGGT; backward: cagtaatacgactcactatagggagaaggctTAAAATTTAAATCCAACTTCCAAAC) for this region were designed using Epidesigner. Our target region contained 25 CpG sites (Table 2); 23 sites could be measured, except CpG 5 and CpG 21, because their fragment molecular weights exceeded the detectable range. CpG 4 and CpG 18 had the same mass spectrometric data because their molecular weights were the same, expressed as CpG 4/18, similar to CpG 8/12/24. The molecular weight of CpG 9-, CpG 10-, and CpG 11- containing fragments overlapped, thus, the average of the three CG-containing fragments represented DNA methylation of CpG 9.10.11, similar to CpG 16.17 and CpG 19.20. However, CpG 1, 3, and 19.20 were not detected in some animals, so no statistical analysis was performed.

### 2.5. Statistical Analysis

The CAD (total capillary area/tissue area), CSD (total capillary circumference/tissue area), CND (total number of capillaries/tissue area), and APC (area per capillary) of the caruncle were measured and calculated using Sunny Digital microscope lab system computer software (Ningbo Sunny Instruments Co., Ltd., Ningbo, China). All the results from the individual groups were presented as the mean ± standard error of mean (SEM). The statistical evaluation was determined by one-way ANOVA, combined with the unpaired two-tailed Student’s *t*-test, at each time point for all the groups using GraphPad Prism 5.0 software (GraphPad Software, San Diego, CA, USA). ^a, b, c, d^ *p* < 0.05 was considered statistically significant. We used SPSS 24.0 Bivariate Correlations (IBM, Armonk, NY, USA) for correlation analysis and *p* < 0.05 indicated significant r^2^ values.

## 3. Results

CD31 staining to label endothelial cells in the caruncle tissue revealed an extensive vascular distribution from day 20 to day 60 of pregnancy, but the enlarged vessels were evident on day 60 of pregnancy (Figure 1A–E). Proliferating cells were marked using a proliferating cell nuclear antibody (Ki67; Figure 1F–J), and Ki67-positive cells were detected in the caruncle between day 20 and day 60 of pregnancy.

Figure 2 shows the patterns of changes for the vascular measurements, including CAD, CSD, CND, APC, and Ki67 in the caruncle. CAD increased gradually during the early stage of pregnancy; On day 60, it was 1.5 to 2.1-fold greater, compared with days 20–45 of pregnancy (*p* < 0.007) (Figure 2A). CSD increased between day 30 and day 60 of pregnancy (*p* < 0.03) (Figure 2B). Changes in CND were insignificant between days 20 and 60 of pregnancy, probably due to the variability between animals (Figure 2C). On day 60, there was a 3.7-fold increase in APC, compared with day 30 of pregnancy (*p* < 0.05) (Figure 2D). The rate of proliferation of endothelial and perivascular cells increased gradually with the progress of the pregnancy. On days 45 and 60, there was a 1.7- to 2-fold increase in the ratio of Ki67-positive cells, compared with day 20 of pregnancy (*p* < 0.002) (Figure 2E). Additionally, the ratio of Ki67-positive cells was positively correlated with CAD (r^2^ = 0.726; *p* < 0.003).

Figure 3A–H show the mRNA expression of *VEGFA*, *FLT1*, *KDR*, *ANGPT1*, *ANGPT2*, *TEK*, *FGF2* and *FGFR2*. The mRNA expression of *VEGFA* was lowest on day 30, which was 3.9- to 5-fold lower than on days 45–60 of pregnancy (*p* < 0.02) (Figure 3A). However, the mRNA expression of *FLT1* was highest on day 30, which was 2.2- to 4.3-fold higher than on days 20–25 and 45–60 of pregnancy (*p* < 0.05) (Figure 3B). The mRNA expression of *KDR* was 2- to 2.5-fold higher on days 30 than on days 20 and 45–60 of pregnancy (*p* < 0.02) (Figure 3C). The mRNA expression of *ANGPT1* increased by 2- to 9.6-fold on day 30, compared with days 20–25 and 45–60 of pregnancy (*p* < 0.02) (Figure 3D). The mRNA expression of *ANGPT2* increased by 1.5- to 3.3-fold between days 20 and 45, but decreased by day 60 of pregnancy (*p* < 0.05) (Figure 3E). The mRNA expression of *TEK* approximately increased by 2.5- to 5.6-fold on day 25, compared with days 20 and 30–60 of pregnancy (*p* < 0.01) (Figure 3F). The mRNA expression of *FGF2* decreased gradually during the early stage of pregnancy and was very low on days 45–60 (*p* < 0.05) (Figure 3G). The mRNA expression of *FGFR2* was approximately 2.7- to 12.3-fold higher on day 20 than on any other day of pregnancy (*p* < 0.01) (Figure 3H).

There was a negative correlation between CAD and *FGF2* mRNA expression (r^2^ = −0.716; *p* < 0.004). CSD was negatively correlated with mRNA expression of *TEK* (r^2^ = −0.625; *p* < 0.02) and *FGF2* (r^2^ = −0.648; *p* < 0.01). CND was positively correlated with mRNA expression of *ANGPT2* (r^2^ = 0.641; *p* < 0.02). APC was negatively correlated with mRNA expression of *ANGPT2* (r^2^ = −0.589; *p* < 0.03). A negative correlation was observed between the ratio of Ki67-positive cells and mRNA expression of *FGF2* (r^2^ = −0.707; *p* < 0.004), and *FGFR2* (r^2^ = −0.741; *p* < 0.003) (Table 3).

There was a decrease in the mean methylation of the *ANGPT2* on days 45 and 60, compared with day 25 of pregnancy (*p* < 0.05) (Figure 4A), which was also found to be negatively correlated with CAD (r^2^ = −0.563; *p* < 0.004). There was a decrease in methylation (a) at the CpG 4/18 site on day 60, compared with days 25 and 30 of pregnancy (*p* < 0.04) (Figure 4C); (b) at the CpG 9.10.11 site on day 60, compared with day 30 of pregnancy (*p* < 0.02) (Figure 4G); (c) at the CpG 15 site on day 60, compared with days 30 and 45 of pregnancy (*p* < 0.05) (Figure 4J). However, DNA methylation levels of other CpG sites did not display a significant difference among day 20 to 60 of pregnancy (*p* > 0.05).

CAD was negatively correlated with methylation at the CpG 6 (r^2^ = −0.565; *p* < 0.04), CpG 7 (r^2^ = −0.545; *p* < 0.05), and CpG 15 (r^2^ = −0.572; *p* < 0.04) sites. CSD was negatively correlated with methylation at the CpG 2 (r^2^ = −0.654; *p* < 0.02) and CpG 16.17 (r^2^ = −0.676; *p* < 0.009) sites. CND was positively correlated with methylation at the CpG 15 (r^2^ = 0.678; *p* < 0.01) site. APC was negatively correlated with methylation at the CpG 15 (r^2^ = −0.873; *p* < 0.0002) site. Additionally, the mRNA expression of *ANGPT2* was positively correlated with methylation at the CpG 15 (r^2^ = 0.594; *p* < 0.03) site, and negatively correlated with the CpG 23 site (r^2^ = −0.534; *p* < 0.05) (Table 4).

## 4. Discussion

In the present study, we evaluated the vascular development and endothelial cell proliferation in the caruncle, and their relationship with the abundance of major angiogenic factor transcripts during the early stage of pregnancy in goats. Our research demonstrated that the proportion of proliferating cells was observed to be very high (>26%), and increased approximately 2-fold from days 20 to 60 of pregnancy. The high proliferation rates were consistent with the results in humans, marmosets, pigs, sheep, and rats during the early stage of pregnancy [5,24,25,26,27]. These data suggested that the proliferated cells provided a cellular basis for the maternal recognition of pregnancy, initial implantation, and angiogenesis. Here, we found that blood vessels marked with the CD31 antibody were detected in the caruncle from day 20 to day 60 of pregnancy, but from day 25 to day 60 in the fetal membranes (unpublished observations). The CAD increased gradually as the pregnancy progressed, which was positively correlated with the proportion of proliferating cells. Previous studies have confirmed that proliferating cells contribute to angiogenesis, which was reflected by an increase in CAD, CSD, and APC in the endometrium [5]. In this study, the CND increased while the APC remained unchanged between days 20 and 30 of pregnancy; additionally, the CND decreased and the APC increased during on days 30–60 of pregnancy. Our previous studies have shown that fetal membranes invade the caruncle and initiate the formation of cotyledons (with some blood vessels) on day 30 of pregnancy [28]. Based on these results, it was reasonable to postulate that the appearance of cotyledons could change the pattern of placental angiogenesis. However, the molecular mechanism of regulating caruncle angiogenesis in goats needs further investigation.

The VEGF and ANGPT family are two specific signaling pathways, which regulate vascular endothelial cell survival, migration, proliferation, and vascular remodeling in the placenta [29,30,31]. In this study, we demonstrated that blood vessels are initiated very early in the caruncle, since the expression of major angiogenic factors was maintained at a relatively high level between days 20–25 of pregnancy. However, the expression of *VEGFA* was lower on day 30 of pregnancy, but the expression of *FLT1*, *KDR*, *ANGPT1,* and *ANGPT2* was higher on day 30 of pregnancy. This data indicated that other members of the VEGF family, such as the *PlGF* gene [30] and *ANGPTs,* might compensate for the low expression of *VEGFA*. Similarly, *FGF2* and its receptors could also play a compensatory role in angiogenesis [32]. Furthermore, the changes in the mRNA expression of *ANGPT1* was consistent with *TEK*, based on a previous study that suggested that ANGPT1 could promote the early placental development by binding to TEK in blood vessels through a paracrine mechanism [33]. We observed increases in *ANGPT2* mRNA expression from day 20 through to day 45 of pregnancy for the caruncle. Interestingly, the expression of *ANGPT2* was positively correlated with CND, which suggested that ANGPT2 increased the area of blood vessels by affecting the number of blood vessels during the early stage of pregnancy. Fagiani et al. found that ANGPT2 participated in vascular remodeling through the autocrine pathway [34].

In this study, there was a 2- to 3-fold decrease in the expression of *FGF2* and *FGFR2* in the caruncle between days 20 and30 of pregnancy. Similar results, showing a weakened expression of *FGF2* and its receptor in the porcine endometrium from day 10 of pregnancy, were reported by Welter et al.; this indicated that FGF2 was related to the formation of blood vessels in early pregnancy [35,36]. From days 45 to 60 of pregnancy, the expression of *FGF2* and *FGFR2* remained low and changed little, similar to the results from days 50 to 140 of pregnancy in sheep maternal placenta [1]. These data further indicated that *FGF2* and *FGFR2* might have a role in the regulation of angiogenesis during the initial stage of pregnancy. Furthermore, inhibition of *FGFR2* expression caused a decrease in the formation of trophoblast and limited trophoblast outgrowth [37,38,39]. In humans, upregulated FGF2 signaling pathways are known to promote placental artery endothelial cell proliferation and angiogenesis [16,17,18]. However, in this study, the expression of *FGF2* was negatively correlated with the proportion of proliferating cells and CAD. It is speculated that the reason could be the difference in the physiology of a goat placenta compared with a human placenta, but the mechanism needs further investigation.

DNA methylation is a well-known epigenetic mechanism that is involved in gene regulation. In an abnormal uterus, there is a modified pattern of DNA methylation, which further alters gene expression and affects uterine development [40]. Previous studies have shown an inverse relationship between the methylation of *VEGF* promoter and the levels of *VEGF* mRNA; this indicates that the upregulation of *VEGF* was beneficial to maintaining normal angiogenesis in preterm preeclampsia [41]. This data indicates that DNA methylation plays an important role in the regulation of placental angiogenesis, but the studies performed during the early stage of placental development are scarce.

ANGPT2 is a growth factor secreted by vascular endothelium, which can enhance the activity of endothelial cells, maintain vascular structure, and assist endothelial cells to migrate and proliferate, thus promoting angiogenesis [10,11]. In this study, the mean methylation of ANGPT2 was negatively correlated to the expression of ANGPT2. This result was consistent with the results in chronic lymphocytic leukemia [42]. Moreover, the mean methylation of ANGPT2 was also negatively correlated with CAD, which indicated that modification by DNA methylation could affect angiogenesis by regulating *ANGPT2* expression. Furthermore, several CpG sites in the target region of these genes showed differential methylation during early pregnancy, similar to *VEGF* in preeclampsia [41]. Interestingly, the methylation at the CpG 15 site was hypermethylated, and significantly correlated with CAD, CND, APC, and the expression of *ANGPT2*. Therefore, our findings suggested that DNA methylation of CpG 15 at *ANGPT2* could play an important role in caruncle angiogenesis during the early stage of pregnancy, but the methylation modification needs further investigation.

## 5. Conclusions

The present study shows that gene expression mediates the association between the DNA methylation of *ANGPT2* and angiogenesis in the caruncle during the early stage of pregnancy. Our findings support the importance of placental epigenetic changes in placental angiogenesis and development. Future work should focus on investigating how the DNA methylation of *ANGPT2* modulates angiogenesis in the caruncle during the early stage of pregnancy.

## Figures and Tables

**Figure 1 animals-13-00099-f001:**
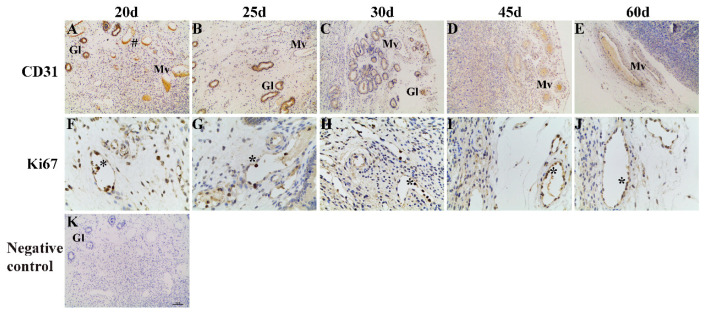
Representative photomicrographs of immunohistochemical staining for CD31 and Ki67 in the goat caruncle. (**A**–**E**) 100×. Apparent immunoreactive CD31-positive vessels labeled on days 20 (**A**), 25 (**B**), 30 (**C**), 45 (**D**), and 60 (**E**) of pregnancy. (**F**–**J**) 400×. Higher density of Ki67 (black arrows) in the nuclei of endothelial and perivascular cells from goats on days 20 (**F**), 25 (**G**), 30 (**H**), 45 (**I**) and 60 (**J**) of pregnancy. (**K**) 100×. Negative control in caruncle on day 20 of pregnancy; no staining was observed. Pound sign indicates staining of CD31 in the caruncle; Mv = maternal vessels; Gl = uterine glands; Asterisks indicate staining of Ki67 in the endothelial and perivascular cells of the caruncle.

**Figure 2 animals-13-00099-f002:**
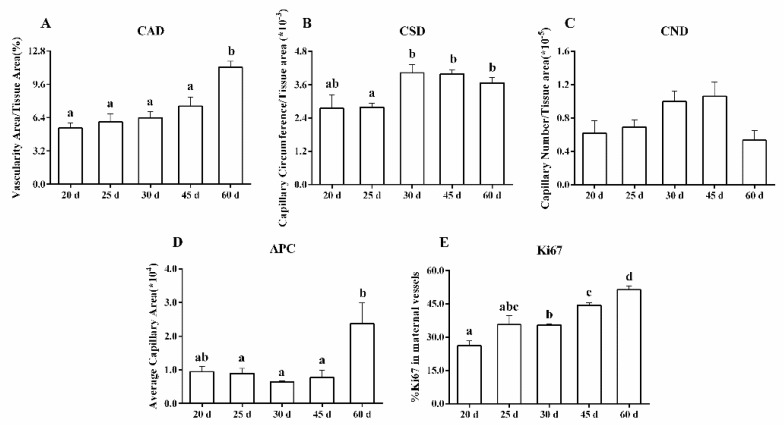
Measurements of vascularity including the capillary area density (CAD; (**A**)), the capillary surface density (CSD; (**B**)), the capillary number density (CND; (**C**)), the area per capillary (APC; (**D**)), and the proportion of proliferating cells (Ki67; (**E**)), on days 20, 25, 30, 45, and 60 of pregnancy in the caruncle. ^a, b, c, d^ *p* < 0.002–0.05; values ± S.E.M. with different superscripts differ within specific measurement.

**Figure 3 animals-13-00099-f003:**
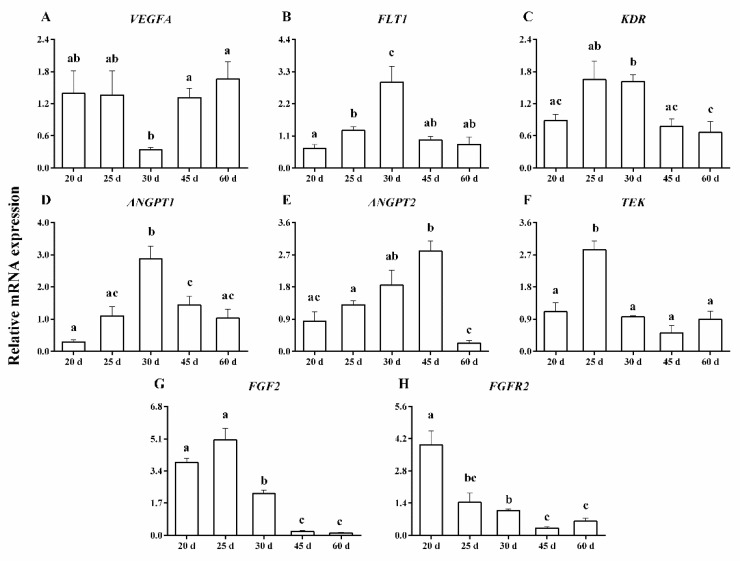
The mRNA expression of the *VEGFA* (**A**), *FLT1* (**B**), *KDR* (**C**), *ANGPT1* (**D**), *ANGPT2* (**E**), *TEK* (**F**), *FGF2* (**G**), and *FGFR2* (**H**) in the caruncle on days 20, 25, 30, 45, and 60 of pregnancy. ^a, b, c^ *p* < 0.01–0.05; values ± S.E.M. with different superscripts differ within each specific gene.

**Figure 4 animals-13-00099-f004:**
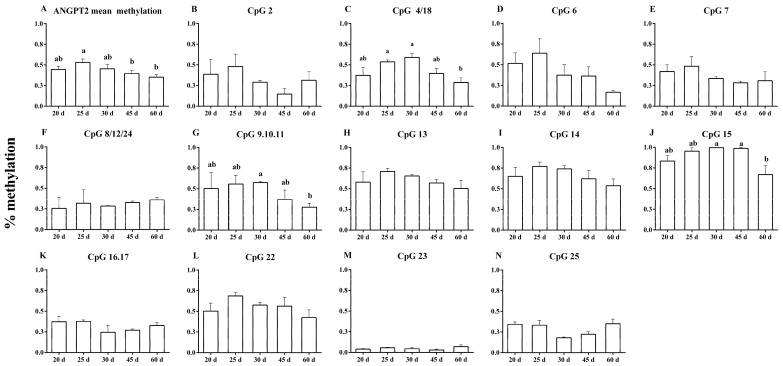
Mean methylation levels and differentially methylated sites in the *ANGPT2* target region in the caruncle on days 20, 25, 30, 45, and 60 of pregnancy. *ANGPT2* mean methylation levels (**A**); CpG 2 (**B**); CpG 4/18 (**C**); CpG 6 (**D**); CpG 7 (**E**); CpG 8/12/24 (**F**); CpG 9.10.11 (**G**); CpG 13 (**H**); CpG 14 (**I**); CpG 15 (**J**); CpG 16.17 (**K**); CpG 22 (**L**); CpG 23 (**M**); and CpG 25 (**N**). ^a, b^ *p* < 0.02–0.05; values ± S.E.M. with different superscripts differ within each specific gene.

**Table 1 animals-13-00099-t001:** The information of primer sequence designed.

Gene	Accession No.	Primer Sequence (5′→3′)	Product Length (bp)
*VEGFA*	XM_018038497.1	F: GTGACCCAGCACAGTTCCTCTT	74
		R: TTCCGGGCTCGGTGATTTAG	
*FLT1*	XM_018056600.1	F: TGCCAGCAAGTGGGAGTTT	102
		R: TTGATGCCGAATGCCGATG	
*KDR*	XM_005681629.3	F: TCAACAGGATGGCAAAGACT	96
		R: GAAACAGGTGAGGTAGGCAGA	
*ANGPT1*	XM_ 005689163.3	F: CTGGTGGTTTGATGCCTGTG	182
		R: GCTCTTGATTGCTATTGTGCC	
*ANGPT2*	XM_013975359.2	F: GACTGGTCAGAAACGCTACG	168
		R: GGCTGGCTTATGCTGCTTATT	
*TEK*	XM_018051890.1	F: GGGGAAAGGTATGAGTGCTAA	117
		R: AGGATGGCTTCTACAGGTGAG	
*FGF2*	XM_018061205.1	F: GCAAACCGTTACCTTGCTAT	153
		R: TCGTTTCAGTGCCACATACC	
*FGFR2*	XM_018041441.1	F: ACTGTGGTTGGAGGCGATGT	105
		R: CCCGTATTTACTGCCGTTCTT	
β-actin	NM_001314342.1	F: TGATATTGCTGCGCTCGTGGT	189
		R: GTCAGGATGCCTCTCTTGCTC	

**Table 2 animals-13-00099-t002:** PCR primers for *ANGPT2* gene region.

Chromosomal Region	Chr27:39007363 to 39007851
Target sequence (3′→5′)	ggagtaagtatttcaggttcctgggccacaC^[1]^GggctctgtcacagctgctC^[2]^GactctgcC^[3]^GttgtagtgtagaagccatcaagagaaaaccctC^[4]^GaggatgggaggtggctgtgtgccaaC^[5]^GaagcC^[6]^GtattC^[7]^GttgacactgaaatttgaacttatcataatttccaC ^[8]^GtcatgaaatattctctggatttttttcagtcttttaaaaatgtaggaaccattcttagctcactggcC^[9]^GttcC^[10]^GC^[11]^GgggacC^[12]^GcaggccctggcctgC^[13]^GggctggagggggttgC^[14]^GggggggctgtgacaC^[15]^GgtctccaggcctctcC^[16]^GggC^[17]^GcaggcagtgtaagagggaacctccaggagccttccaC^[18]^GcctcagagcagC^[19]^GccC^[20]^GttctcccaaggaC^[21]^GagaactgtgagacaggcC^[22]^GggggaaagggctcattgaagggaC^[23]^GggcacattC^[24]^GaccaaaaaggaaagtgcC^[25]^Gcttggaagctggatctaaactcca
location	5′UTR; N-shore
Primers (5′→3′)	Forward: aggaagagagGGAGTAAGTATTTTAGGTTTTTGGGT Reverse: cagtaatacgactcactatagggagaaggctTAAAATTTAAATCCAACTTCCAAAC
Amplicon length (bp)	489

Grey shading CpG sites cannot be detected by Sequenom MassARRAY.

**Table 3 animals-13-00099-t003:** Correlation of vascular distribution with the mRNA expression of the major angiogenic factors in the caruncle during the early stage of pregnancy.

Angiogenic Factor		Ki67	CAD	CSD	CND	APC
*VEGFA*	r^2^	0.260	0.138	−0.167	−0.131	0.117
	*p*	0.350	0.624	0.552	0.641	0.678
*FLT1*	r^2^	−0.174	−0.222	0.396	0.378	−0.398
	*p*	0.535	0.427	0.144	0.165	0.142
*KDR*	r^2^	−0.274	−0.474	−0.074	0.258	−0.505
	*p*	0.322	0.074	0.794	0.352	0.055
*ANGPT1*	r^2^	0.063	−0.092	0.469	0.437	−0.265
	*p*	0.823	0.744	0.077	0.103	0.339
*ANGPT2*	r^2^	0.004	−0.404	0.308	**0.641**	**−0.589**
	*p*	0.988	0.135	0.265	**0.010**	**0.021**
*TEK*	r^2^	−0.231	−0.377	**−0.625**	−0.288	−0.077
	*p*	0.408	0.165	**0.013**	0.299	0.784
*FGF2*	r^2^	**−0.707**	**−0.716**	**−0.648**	−0.189	−0.399
	*p*	**0.003**	**0.003**	**0.009**	0.501	0.140
*FGFR2*	r^2^	**−0.741**	−0.505	−0.484	−0.238	−0.193
	*p*	**0.002**	0.055	0.067	0.393	0.492

Values in bold indicate significant r^2^ values.

**Table 4 animals-13-00099-t004:** Correlation of vascular distribution with the DNA methylation of CpG sites at *ANGPT2* in the caruncle during the early stage of pregnancy.

CpG Sites		CAD	CSD	CND	APC	*ANGPT2*
CpG 2	r^2^	−0.361	**−0.654**	−0.464	−0.071	−0.325
	*p*	0.205	**0.011**	0.095	0.809	0.256
CpG 4/18	r^2^	−0.505	−0.155	0.189	−0.508	0.231
	*p*	0.066	0.598	0.517	0.064	0.427
CpG 6	r^2^	**−0.565**	−0.342	−0.023	−0.421	0.191
	*p*	**0.035**	0.232	0.937	0.134	0.513
CpG 7	r^2^	**−0.545**	−0.481	−0.095	−0.413	−0.187
	*p*	**0.044**	0.082	0.746	0.142	0.522
CpG 8/12/24	r^2^	−0.036	−0.068	0.011	0.040	−0.087
	*p*	0.902	0.816	0.970	0.892	0.768
CpG 9.10.11	r^2^	−0.517	−0.495	−0.233	−0.310	0.038
	*p*	0.059	0.072	0.424	0.280	0.897
CpG 13	r^2^	−0.523	−0.393	−0.036	−0.443	0.108
	*p*	0.055	0.165	0.902	0.113	0.713
CpG 14	r^2^	−0.532	−0.388	−0.088	−0.440	0.108
	*p*	0.050	0.171	0.765	0.115	0.713
CpG 15	r^2^	**−0.572**	0.184	**0.678**	**−0.873**	**0.594**
	*p*	**0.033**	0.529	**0.008**	**0.000**	**0.025**
CpG 16.17	r^2^	−0.250	**−0.676**	−0.417	0.008	−0.520
	*p*	0.388	**0.008**	0.138	0.978	0.057
CpG 22	r^2^	−0.417	−0.255	0.014	−0.427	0.161
	*p*	0.138	0.379	0.963	0.128	0.582
CpG 23	r^2^	0.248	−0.051	−0.133	0.422	**−0.534**
	*p*	0.393	0.863	0.650	0.133	**0.049**
CpG 25	r^2^	−0.059	−0.484	−0.439	0.107	−0.485
	*p*	0.842	0.079	0.117	0.717	0.079

Values in bold indicate significant r^2^ values.

## Data Availability

The datasets used and/or analyzed during the current study are available from the corresponding author on reasonable request.

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
