# Peer review of "Vascular Distribution and Expression Patterns of Angiogenic Factors in Caruncle during the Early Stage of Pregnancy in Goats (Capra hircus)"

_animals, 2022, doi:10.3390/ani13010099_

Round 1

Reviewer 1 Report

The authors investigated the vascular distribution, mRNA expression of major angiogenic factors, and the methylation levels of ANGPT2 in the goat caruncle. I have several comments to improve this manuscript.

1. Please add the detail of RT-PCR reaction system in the part of material and method.

2. The A-E magnification in Figure 1 is 100×,  however, F-J is 400×, K is 100×. A-E and F-J use the same control group, and it is recommended to use pictures with the same magnification. In addition, add the corresponding control pictures after 25, 30, 45 and 60 day .

3. There are some mistakes in the significance marking in Figure 2 and Figure 3. Please calculate positive rate of CD31 protein in Figure 2.

4. The qulity of all figures need to be improved and please modified the fomat of tables 3 and 4.

5. Whether the methylation level of FGF2 and FGFR2 genes is related to the phenotype.

Reviewer 2 Report

Dear authors,
I suggest some improvements to your manuscript as follows:
1) I suggest that the title of the manuscript be more precise and clear, and that the genes studied (abbreviations) be listed instead of "key genes".
2) The literature review needs to describe in more detail the current state of scientific knowledge about the "key genes" whose activity is being studied. It is necessary to inform the reader in more detail about the research questions.
3) In the Materials and Methods chapter, the biological sample, the number of goats, the age of the goats, and the breed of goat, if known, need to be described in more detail.
4) I suggest that the statistics be described in more detail.
5)The literature in the reference list must be written according to the instructions of the Animals journal.

Reviewer 3 Report

Dear Authors, 

The manuscript is of great significance in understanding early gestation and placentation in goats. 

The introduction lacks some explanation about the target genes, which is given in the discussion. This reviewer thinks that the first sentence of each top in the discussion, which introduces what that gene is about, must be in the introduction.

M&M:

Lines 83 and 84:  it was not clear in which areas of the uterus the tissue was collected if it was close to the embryo or in the uterine body. Generally, embryos grow in the uterine horns. If the authors collected both, I think it would be reasonable to compare these two areas to understand if they are any different.

Line 101: I did not understand why the authors used 2-3 areas of the uterine section/sheep. And now I have four doubts: 1. Was it sheep or goat? 2. How many caruncles of each goat/area were collected? Were all the collected material analyzed? 4. Which area of the caruncles (edge or center) were analyzed?

Results

Te results are well presented and te figures help the readers in understanding all of it. 

Discussion

The first paragraph is unnecessary and looks like an introduction and not a discussion. As in the first sentence of each theme in the discussion, as said before. I missed more discussion about the target genes and their expression through time, the changes that happened.
